# Leveraging Trusted Execution Environments For Data Security in Healthcare Workflows

Chitrabhanu Gupta[†], Chen-Nee Chuah[†], Sean Peisert[§], Venkatesh Akella[†]

[†]Department of Electrical and Computer Engineering, University of California, Davis, USA
Email: {cbgupta, chuah, akella}@ucdavis.edu
[§]Lawrence Berkeley National Laboratory, USA
Email: sppeisert@lbl.gov

*Abstract*—Modern biomedical AI pipelines require robust data protection across heterogeneous environments, including edge devices, hospital servers, and cloud resources, each with distinct performance, trust, and regulatory considerations. While recent advancements in hardware-backed confidential computing (e.g., Intel SGX, AMD SEV, ARM TrustZone) offer promising solutions for data security, their differing threat models prevent seamless, end-to-end "capture-to-use" protection. To address this, we propose a novel, hardware-agnostic security monitor that extends the attestation and memory-encryption capabilities of these disparate Trusted Execution Environments (TEEs). This is complemented by a software-defined secure tunnel that enforces data-centric policy, provenance, and compliance. Our proof-of-concept prototype, integrating a TrustZone-enabled Raspberry Pi with an AMD SEV virtual machine in a cloud environment, demonstrates a deployable, data-centric enclave architecture that achieves end-to-end confidentiality, integrity, and compliance without compromising clinical throughput in biomedical AI workflows.

*Index Terms*—privacy preserving workflows, data privacy security in healthcare, confidential computing

## I. INTRODUCTION

Biomedical sensing platforms now generate continuous streams of high-dimensional data, including radiological images, electrophysiological waveforms, and ultrasound video. These data feed an increasingly diverse set of machine-learning (ML) models that assist in diagnosis, therapy planning, and long-term population studies. The clinical appeal of such systems is well established: they offer early detection, improved prognostication, and potential reductions in clinician workload. Nevertheless, the infrastructure that underpins many pipelines is assembled in an ad-hoc manner, constrained by local expertise, budgets, and privacy regulations such as HIPAA, GDPR, and regional Institutional Review Board (IRB) policies. As a result, three systemic limitations persist.

First, collaborative access to multimodal datasets remains cumbersome; transferring protected health information (PHI) across institutional boundaries often requires months of legal review. Second, scalability is uneven: workloads that begin compute-dominant during training can become I/O-bound during large-scale inference or vice-versa, forcing repeated re-engineering of storage and networking layers. Third, traditional security controls enlarge both the trusted computing base (TCB) and the "trusted people base," exposing attack surfaces that extend from bedside devices to cloud services and administrative personnel.

Recent trusted execution environments (TEEs)—ARM TrustZone, RISC-V Keystone, Intel SGX/TDX, AMD SEV, and NVIDIA Confidential Computing Architecture—offer hardware-enforced isolation, memory encryption, and remote attestation. When deployed along the sensor–edge–cloud continuum, these primitives can deliver uniform confidentiality and integrity guarantees while preserving the elasticity of modern cloud platforms. Moreover, the audit logs generated by TEEs simplify proof-of-compliance, easing regulatory certification and incident forensics. Despite these advantages, no single TEE implementation spans all device classes found in biomedical settings; data must traverse heterogeneous enclaves with differing threat models, resource budgets, and latency profiles. Coordinating keys, attestation formats, and secure transport across this heterogeneity remains an open engineering challenge.

To ground the discussion, we analyze a representative acute respiratory distress syndrome (ARDS) detection workflow. Mechanical ventilators in the intensive-care unit record high-frequency pressure and flow waveforms; a previously published pilot study [1] of an AI-driven clinical decision support system employs Raspberry Pi to aggregate and stream these data to a central server that executes ML models and returns bedside alerts. Clinicians label events via an iOS application, creating a continuously growing dataset. This pipeline embodies the heterogeneity described above: resource-constrained edge devices, a hospital-owned compute cluster, and external research servers that require de-identified subsets of the data. On the basis of this analysis, we propose a composable, data-centric enclave architecture that (i) unifies attestation and transport across heterogeneous TEEs, (ii) reduces the trusted hardware and personnel bases, and (iii) imposes negligible latency overhead on real-time inference. A prototype combining ARM TrustZone at the edge and AMD SEV in the cloud demonstrates the practicality of the approach.

This paper makes two key contributions. First, we identify limitations of current off-the-shelf Trusted Execution Environments (TEEs) in establishing end-to-end data pipelines for healthcare applications. We then propose software techniques to overcome these limitations and align threat models. Second,

we demonstrate how to create software-defined secure data tunnels that augment existing TEEs. These tunnels ensure data-centric policy, compliance, and provenance guarantees, which are crucial for data privacy in biomedical research and healthcare.

The remainder of the paper is organized as follows. Section II reviews common biomedical data modalities and details the ARDS detection pipeline. Section III evaluates the security vulnerabilities present in current workflows. Section IV introduces the enclave-based design, and Section V describes a software-defined secure data tunnel that provides end-to-end provenance and policy enforcement. Section VI discusses related work, and Section VII concludes.

## II. BACKGROUND

Clinical practice routinely generates data in three broad modalities—images, waveforms, and video—each with well-studied machine-learning (ML) applications. Radiological and microscopic images underpin models for Alzheimer's staging, lung and breast cancer detection, COVID-19 diagnosis, and vasculature segmentation [2]–[7]. Electrophysiological time series such as EEG and ECG support arrhythmia prediction, schizophrenia screening, and emotion recognition [8]–[13]. Continuous video, captured by ultrasound or endoscopic cameras, enables automated assessment of fetal cardiac structure [14] and progression of aortic stenosis [15].

While these tasks differ clinically, their computational pipelines share a common skeleton: a sensor acquires raw data, an edge device performs lightweight processing, a central server stores and analyses the data, and peripheral nodes conduct additional experiments or anonymization. The same outline scales from small pilot studies to national biobanks; differences arise mainly in throughput, latency tolerance, and regulatory constraints.

### A. Motivating Workflow: ARDS Detection

To ground the discussion, we adopt the acute respiratory distress syndrome (ARDS) detection pipeline described by Rehm *et al.* [1], [16]. Mechanical ventilators in an intensive-care unit record high-frequency pressure and flow waveforms. A Raspberry Pi running Linux aggregates the signals, executes minimal preprocessing, and streams the data over a secure wireless link to a hospital server. Clinicians label each time series via an iOS application, pairing waveform segments with electronic health record (EHR) metadata to create a supervised training set for ML models. Although modern mechanical ventilators deployed in major hospitals may support seamless logging and streaming of data without support from a separate embedded device, the most advanced technology is often unobtainable for resource constrained clinics around the globe. Thus, the pipeline from this pilot study still remains relevant to AI-driven healthcare delivery across disease domains.

In this workflow, the waveforms themselves are sensitive: patient identifiers can be linked, via timestamps, to the hospital's EPIC EHR system. Consequently, researchers external to the hospital must not receive raw data. A hospital-employed data engineer anonymizes and filters the server's dataset, exporting de-identified subsets to a relational database (RDBMS) hosted on a separate server. Researchers issue SQL queries for exploratory statistics, then copy the resulting tables to local workstations or lab-scale clusters for compute-intensive training runs.

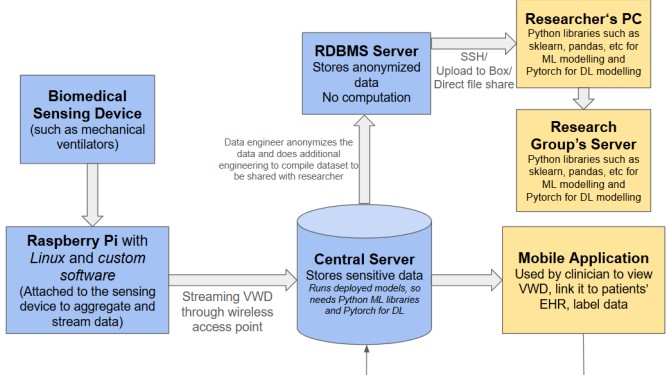

Fig. 1. ARDS Detection Pipeline. Ventilator waveforms travel from edge device to hospital server, are anonymized, and are later accessed by external researchers for model development.

Figure 1 illustrates the full data path. Every arrow represents a potential disclosure or tampering point; every storage node must satisfy encryption-at-rest requirements; and every user group—clinicians, data engineers, and researchers—adds to the trusted personnel base.

### B. Large-Volume Video Workflows

Some health-AI studies accumulate far larger datasets than the ARDS example. Wu *et al.* [17] analyze 1707 three-minute parent–infant interaction videos ($\approx 200$ GB) to predict early autism risk. Even this "modest" corpus was judged insufficient for deep neural networks, indicating that future deployments will involve terabyte-scale video repositories. Because PHI cannot leave hospital premises without extensive legal review, researchers typically ship gradients or model updates to cloud nodes (federated learning) while retaining raw video on-site. Thus, although topologies vary, the security and privacy requirements closely mirror those of the ARDS pipeline: strong isolation at the edge, encrypted transport, auditability, and controlled external access.

These examples motivate the design goals pursued in the remainder of this paper: end-to-end confidentiality, interoperability across heterogeneous devices, and minimal performance overhead.

## III. VULNERABILITIES IN HEALTH AI WORKFLOWS

Every stage of the ARDS detection pipeline exposes security weaknesses. At the edge, the Raspberry Pi attached to the ventilator is physically reachable by patients, visitors, and clinical staff; an adversary with brief access can replace the micro-SD card, attach a rogue storage device, or reboot the unit into a compromised image. Wireless telemetry between

the Pi and the hospital server is sent without end-to-end encryption, permitting passive interception or active manipulation in transit.

On the hospital server, raw waveforms and patient identifiers share the same software stack as the annotation interface and the inference service. The absence of memory encryption, process isolation, and hardened hypervisor protections means that a single system compromise reveals the complete dataset. Data engineers with shell access can inadvertently copy sensitive records to an RDBMS intended only for de-identified tables; once present on that auxiliary server, the data is less rigorously monitored and may propagate to external research machines. Those downstream PCs and lab servers operate outside hospital governance, often without full-disk encryption or consistent patch management.

The mobile application used by clinicians inherits the privileges of the server it contacts; a vulnerability in the app or on the iOS device can therefore relay forged labels or extract protected health information (PHI). In aggregate the workflow depends on a broad trusted computing base—edge devices, server OS and hypervisor, mobile OS, personal workstations—and an equally broad trusted personnel base encompassing clinicians, data engineers, researchers, and even visitors.

The autism-risk video pipeline shares these weaknesses but introduces an additional threat: large, centralized video repositories are attractive targets for data-poisoning and model-poisoning attacks. Without fine-grained logging and attestation, malicious content can be injected unnoticed during upload or training, degrading diagnostic accuracy.

### A. Security Requirements of Health AI Workflows

Drawing on the ARDS and autism-risk pipelines, we derive five requirements for a trustworthy biomedical-AI infrastructure:

- **Interoperability.** Pipelines span heterogeneous hardware—from low-power microcontrollers to cloud GPUs—under differing institutional policies. A viable TEE must bridge these environments without ad-hoc gateways or format conversions.
- **Memory encryption and isolated execution.** Each component, including resource-constrained edge devices, needs the option to encrypt at least sensitive memory regions and to confine code handling PHI within an isolated address space.
- **Mutual attestation and integrity verification.** Hardware, firmware, and software must be measured at boot and re-verified before every data exchange; senders and receivers must establish their integrity reciprocally to prevent spoofing.
- **Data-handling compliance.** The infrastructure must tag data with sensitivity and policy metadata, enforce policy during transfer, and generate immutable audit logs to satisfy HIPAA, GDPR, and IRB requirements.
- **Compatibility with complementary privacy technologies.** The TEE should interoperate with secure multi-party computation, differential privacy, and homomorphic encryption, allowing each workflow to combine mechanisms according to latency and accuracy constraints.

## IV. USING EXISTING CONFIDENTIAL COMPUTE TECHNOLOGIES

Sensitive data traverse every stage of the ARDS pipeline bedside acquisition, hospital aggregation, cloud inference, and external research analysis. Deploying commodity trusted-execution environments (TEEs) at each storage or compute node offers isolation, memory encryption, and remote attestation without redesigning the hardware stack.

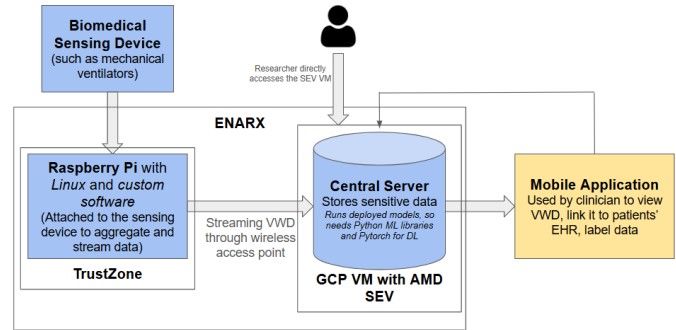

Fig. 2. Proposed ARDS Detection Pipeline

**Raspberry Pi.** The Pi buffers ventilator waveforms and streams them to the hospital network. With ARM TrustZone enabled, the acquisition thread, cryptographic keys, and transmit buffer execute in the *Secure World*; drivers, Wi-Fi, and user shell remain in the *Normal World*. Secure-boot verification and region-based memory protection ensure that PHI never appears in plaintext outside the Secure World. Streaming at 20 kHz incurs 2 % latency relative to an unsecured image well within clinical limits.

**Central server.** Raw data storage, clinician annotation, and real-time inference run on an AMD EPYC VM protected by Secure Encrypted Virtualization Encrypted State (SEV-ES). Page-level memory encryption and register-state protection prevent a malicious hypervisor from inspecting data or model parameters. The Pi initiates data transfer only after verifying the VM attestation report, guaranteeing delivery to a trusted enclave.

**Research RDBMS and compute nodes.** Researchers receive de-identified tables. If policy permits, the database is deployed in a second SEV-ES VM; otherwise it remains on-premises inside an Intel SGX enclave. Result sets are returned over attested TLS channels, so raw identifiers never leave the enclave boundary.

### A. Attestation Framework

Mutual attestation is mandatory before any sensor data move between components. Each TEE generates a signed evidence bundle containing its measurement hash and public key. The sender verifies the receiver's bundle; the receiver repeats the process before accepting the connection. We employ the Enarx

TABLE I
ARDS DETECTION MODELING ON VMS WITH AND WITHOUT SEV.

| Run ID | Regular (s) | SEV (s) |
|--------|-------------|---------|
| 1 | 92.82 | 95.49 |
| 2 | 93.31 | 96.32 |
| 3 | 92.68 | 91.46 |
| 4 | 93.76 | 93.35 |
| 5 | 93.53 | 94.51 |
| 6 | 93.46 | 95.85 |
| 7 | 91.13 | 92.63 |
| 8 | 91.33 | 93.86 |
| 9 | 93.63 | 94.05 |
| 10 | 92.80 | 94.34 |

runtime to normalize these procedures across TrustZone, SEV, and SGX, reducing bespoke code and simplifying future expansions of the pipeline.

### B. Impact of Replacing the Auxiliary RDBMS with SEV

SEV eliminates the need for an intermediate de-identification server and the associated data-engineer role. Researchers analyze data inside the enclave via pre-approved SQL or Python queries whose outputs are automatically scrubbed of identifiers, shrinking both the trusted computing base and the trusted personnel base.

### C. Performance Evaluation

Two Google Cloud VMs one baseline and one SEV-enabled were provisioned with identical resources (2 vCPUs, 8 GB RAM, Ubuntu 20.04). Ten ARDS-model training runs were executed on each VM; Table I lists per-run execution times and means.

The mean increase is 3 %confirming that SEV fulfills latency constraints for bedside alerting.

### D. Limitations of the Proposed Pipeline Due to Interoperability

**Non-uniform guarantees.** TrustZone provides world-level isolation but no full-memory encryption, whereas SEV encrypts guest memory and CPU state. Consequently, security assurances differ between edge and server tiers.

**Hypervisor trust.** SEV assumes an untrusted hypervisor, whereas TrustZone relies on the Non-Secure World OS for certain services. A compromised OS could still launch shared-resource attacks on the Secure World.

**Attestation asymmetry.** SEV ships with built-in remote attestation; TrustZone requires a custom framework (e.g., OP-TEE) to export similar evidence.

### E. Techniques to Mitigate Interoperability

Where TrustZone or other hardware TEEs are unavailable, we encrypt waveforms immediately after capture, containerize edge applications, and perform periodic kernel-hash verification. Although these measures do not equal hardware isolation, they limit exposure until the data reach the SEV enclave. Strict coding standards (constant-time algorithms, minimal attack surface) and noisy execution traces further reduce side-channel risk. Some embedded devices lack stable TrustZone

builds. In that case we retain the SEV-protected server, replace hardware isolation on the Pi with software encryption and signature-based attestation, and omit Enarx, which requires two hardware TEEs. The revised data path is shown in Figure 3.

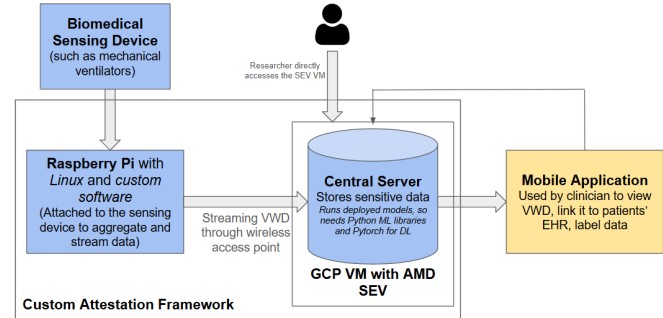

Fig. 3. Proposed ARDS Detection Pipeline Without TrustZone

### F. Threat Model

Table II summarizes common attacks on the minimally protected pipeline (software-only Pi, SEV server) and indicates whether each threat is mitigated, partially mitigated, or unaddressed.

Vulnerabilities due to physical access of devices that could be susceptible to hands-on tampering, malicious insiders, and side-channels are not addressed in this work.

## V. SOFTWARE DEFINED SECURE DATA TUNNEL

Based on these requirements, we outline a hardware-agnostic TEE model suitable for most health-AI pipelines. A thin, high-privilege security monitor executes at the top privilege level on every node. The monitor:

1) orchestrates world or VM switches between secure and non-secure contexts;
2) configures an optional memory-encryption engine, enabling region-level or full-image encryption with ¡ 3 % latency overhead;
3) enforces access control, denying reads or writes to protected regions unless the current context belongs to a registered sensitive application; and
4) provides a uniform attestation API that packages hardware measurements and public keys for remote verification.

Lower-privilege software, including the host OS, lies outside the trusted computing base (TCB) and communicates with the monitor through a narrow driver interface that requires only minor kernel changes.

**Attestation, logging, and policy compliance.** A dedicated compliance server validates attestation reports, signs session keys, and records every data ingress/egress event (timestamp, sender hash, receiver hash, dataset tag). The framework for these operations is built around design principles for attestation and policy compliance from existing literature [18], [19]. If provisioning a standalone server is impractical, the role can be

TABLE II
DEGREE OF PROTECTION AVAILABLE IN MODIFIED PIPELINE AGAINST COMMON VULNERABILITIES

| Attack | Description | Protection Provided |
|---|---|---|
| **Data Interception** | Attackers intercept data in transit between the Raspberry Pi and GCP VM. | Strong Protection: Use of TLS/SSL and application-level encryption ensures data confidentiality. |
| **Memory Dump on GCP VM** | Attackers exploit a vulnerability, accessing memory contents. | Strong Protection: SEV provides memory encryption, protecting sensitive data from being read in memory dumps. In Raspberry Pi, we are implementing encryption as soon as data is generated. |
| **Container Escape** | A malicious process escapes its container, gaining broader access to the host. | Limited Protection: While containerization provides isolation, strict configurations can reduce the risk, but not fully guarantee against escape. |
| **Integrity Verification Failure** | The integrity verification script fails to detect a compromised kernel or application. | Limited Protection: Integrity checks implemented on Raspberry Pi will help, but an advanced attack may bypass them. SEV has its own mechanisms for this |
| **Insider Threat** | Authorized users intentionally or unintentionally expose sensitive data. | Limited Protection: Role-based access controls and auditing can mitigate risks. |

co-located on an existing compute node or distributed across several nodes for fault tolerance. Decentralized attestation removes a single point of failure and provides redundancy when clinical workloads must remain online. Policy rules are stored in a permissions metadata table data structure, which is set by the enclave manager when the application is created. Since this process is handled within the enclave before the OS can intervene, it does not expand the TCB.

**Data-centric enclave workflow.** Each storage or compute node hosts a data-centric enclave. Data move between enclaves only after mutual attestation succeeds and the compliance server issues a signed JSON manifest that contains provenance, policy tags (raw, de-identified, aggregate), and integrity hashes. The receiver's monitor validates the manifest before processing begins, guaranteeing that privacy constraints travel with the data.

This design supports heterogeneous workflows: a central, on-premises video repository (e.g., Wu *et al.* [17]) can expose only approved features, while federated-learning jobs can embed task-level attestation graphs for provenance and auditability as proposed by Guo *et al.* [20]. Granular logging and tagging enable differential policy enforcement—and optional encryption adds a second defense layer—providing continuous protection across diverse clinical AI pipelines.

Components in a workflow that cannot be incorporated into this design will require implementation of a combination of software approximations of security guarantees provided by TEEs and frameworks for privacy preserving technologies. Cloud providers offer substantial engineering support for development and deployment of privacy preserving technologies such as homomorphic encryption, secure multiparty computation and differential privacy. This existing support, combined with the cloud-based design and the uniform attestation and

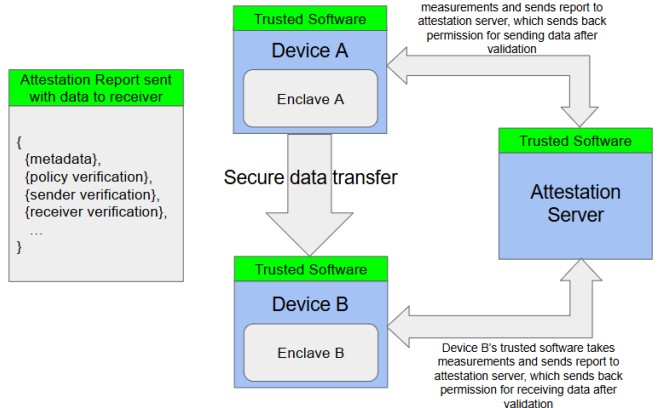

Fig. 4. High-Level Design Schematic of Software Defined Secure Data Tunnel System

policy enforcement protocols, means that setting up the necessary nodes for integrating popular privacy preservation techniques and cross-institutional model training would not require significant engineering effort or any design modifications for the secure tunnel.

Based on our high-level design for a software defined secure data tunnel, we developed a simple prototype to demonstrate that the attestation framework would incur insignificant overhead. We first simulated a scenario where data is transferred from one AMD SEV virtual machine to another, both hosted on Google Cloud Platform. We calculated the time required to attest the identity of the sender and the receiver servers, as well as to check for potential policy violation due to presence of any of the 18 HIPAA identifiers in the data with the help of a dedicated attestation server. We performed the

TABLE III
AVERAGE DURATION FOR COMPLETING THE ATTESTATION PROCESS
BEFORE DATA TRANSFER BETWEEN AMD-SEV SERVERS

| Number of Columns | Average Attestation Time (secs) |
|---|---|
| 10 | 0.0019 |
| 100 | 0.0021 |
| 1000 | 0.0023 |

TABLE IV
AVERAGE DURATION FOR CHECKING THE NATURE OF THE OBJECT TO BE
TRANSFERRED FROM ONE AMD-SEV SERVER TO ANOTHER FOR
FEDERATED LEARNING-LIKE APPLICATIONS

| Type of Object Transferred | Average Validation Time ($\mu$s) |
|---|---|
| Model Updates | 2.6 |
| Raw Data | 2.2 |
| Unrecognized Object | 4.3 |

attestation process for transferred datasets containing 10, 100 and 1000 columns (the number of rows does not matter since the attestation process only requires the metadata), and we report the average attestation time across 5 trials for each different dataset in Table III. The attestation time includes time required to transfer the metadata to the attestation server, perform policy checks and sender and receiver validation, and send back attestation certificates to sender and receiver servers.

We also simulated another scenario in health AI/ML workflows, where operations such as federated learning result in transfer of model updates from one server to another, but raw data is not to be transferred. For instance, the video data for autism risk detection in the workflow described earlier resides at a central repository. Sensitive raw data may not leave the server, but the data might be used for training federated learning schemes for autism risk detection, in which case, data artifacts such as model updates are expected to leave the server, but not the raw video data. Our design supports implementation of policies for such cases, where it needs to be ensured that no raw data leaves the server, and that only a specific type of artifacts are allowed to be transmitted. We perform the attestation process to confirm that raw data is not transferred, only model updates are transferred, and report the overhead incurred for validating that the object being transferred only contains model updates. We report average times required across 5 trials each in case data transfer is attempted, model updates transfer is attempted, and also if some arbitrary unrecognized object transfer is attempted in Table IV.

## VI. RELATED WORK

Confidential computing in healthcare has attracted growing attention. Wu et al. [17] demonstrate the practical challenges of hosting terabyte-scale video datasets on-premises, motivating enclave-based analytics at the edge. Pinto et al. [21] and Costan & Devadas [22] survey ARM TrustZone and Intel SGX, respectively, outlining their suitability for clinical workloads. Several studies integrate SGX with HIPAA-compliant pipelines for genomic privacy [23] and ICU waveforms [24]. Homomorphic-encryption frameworks enable secure model inference but incur order-of-magnitude latency penalties [25]. Secure multiparty computation has been used for cross-institutional cohort analysis [26], yet depends on synchronous, high-bandwidth links uncommon in hospital networks. Differential-privacy defenses mitigate membership inference but trade accuracy for strong bounds [27]. Recent work on enclave-based federated learning adds task-level attestation graphs for provenance auditing [20]. Sherlock, a third-party secure enclave solution, managed by the San Diego Supercomputer Center, provides academic institutions in the U.S. with an opportunity to enter a partnership and obtain a cloud environment rated for compliance with various regulations with managed solutions and platforms for implementation of various security measures [28]. However, such services are typically expensive, and thus often unattainable. Our architecture unifies these strands, combining hardware TEEs with data-centric attestation to deliver end-to-end confidentiality without prohibitive overhead.

## VII. CONCLUSION

We analyzed two representative biomedical-AI pipelines and showed that traditional, ad-hoc security controls broaden both the trusted computing and personnel bases. By deploying commodity TEEs—TrustZone at the edge and SEV/SGX in the cloud and overlaying a software-defined secure tunnel, we reduced the attack surface while adding 3% latency. A thin, hardware-agnostic security monitor plus a dedicated (or distributed) compliance server provides uniform attestation, encryption, and audit logging across heterogeneous devices. Benchmarks confirm that confidential computing now meets bedside timing constraints, eliminating the need for intermediate de-identification servers. Remaining gaps lie in physical tamper resistance and side-channel leakage, which we leave to future work. The proposed data-centric enclave architecture therefore offers a practical path to end-to-end privacy and integrity in next-generation health-AI workflows.

Our design facilitates seamless multi-institution collaboration since it is primarily cloud-based. The standardized metadata table data structure for policy implementation across institutions also makes it easy to scale the attestation and policy implementation framework, and reduces credentialing burden. Institutions are not required to build local infrastructure and can utilize the operational support and flexibility offered by cloud providers that assist development of health-AI applications [29]–[31]. However, some further work is required in order to transform our solution into a production-ready system. we must design an adapter, potentially around a RISC-V based TEE such as Keystone, that interfaces with the output driver of medical devices and sensors and encrypts the data. Additionally, one or more dedicated storage nodes would be set up, which could be supported by AMD SEV. With these additional engineering efforts, it would be possible to deploy a production-ready version of the software defined secure data tunnel.

ACKNOWLEDGMENT

This work was supported in part by Department of Energy ASCR Subaward Cybersecurity for Edge-to-Center Scientific Computing A24-1742 to UC Davis from LBNL and a NSF CNS Core Award 2225882.

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
