# OpenReview forum: "Leveraging Trusted Execution Environments For Data Security in Healthcare Workflows"
_IEEE.org/EMBS/BHI/2025/Conference — BHI 2025_

### Official Review · Reviewer_u87i · 2025-07-14
**Edge-to-cloud pipeline combining TrustZone and SEV secures a medical AI workflow with only about 3 % added latency. Compelling proof of concept, but broader hardware coverage and real-world robustness tests are still needed.**

**Confidence:** 5
**Clarity Of Writing:** great
**Clinical Significance:** good
**Methodological Novelty:** great
**Overall Rating:** 7

**Experiments And Results:**

great

**Questions For The Authors:**

1. Can you include results with SGX or Keystone to confirm the monitor’s vendor neutrality?
2. What safeguards prevent the compliance server from becoming a bottleneck or single point of compromise?
3. How does the system perform over unstable hospital Wi-Fi while meeting bedside alert thresholds?
4. What mechanisms support secure key rotation and enclave updates in offline or low-connectivity environments?
5. Can you provide empirical measurements to show the effectiveness of your proposed side-channel mitigations?
6. How much additional engineering effort would be required to integrate secure multiparty computation for cross-institutional model training?

**Strengths:**

1. Demonstrates an end-to-end architecture that unifies heterogeneous TEEs without ad-hoc de-identification servers.
2. Shows negligible performance cost, confirming that SEV adds little overhead to CPU-bound workloads.
3. Abstracts vendor-specific attestation flows through Enarx, promoting hardware neutrality and simpler maintenance.
4. Embeds data-centric policy tags and immutable audit logging to align with regulatory expectations for health-data security.
5. Provides a thorough threat analysis that spans network interception, insider misuse and container escapes.
6. Acknowledges side-channel issues and proposes practical mitigations such as constant-time code and noise injection.
7. Discusses compatibility with federated learning and differential-privacy extensions, fitting current privacy-preserving trends.
8. Supplies detailed implementation notes and benchmark tables, aiding reproducibility.

**Summary Of The Paper:**

The authors present a security monitor and software-defined tunnel that join ARM TrustZone on a Raspberry Pi edge device with AMD SEV-protected virtual machines in the cloud. The monitor normalises attestation evidence via Enarx, while a compliance server records policy tags and audit logs. In an ARDS-detection case study, encrypted streaming, mutual attestation and provenance tracking raise latency by roughly 3 % and keep attestation exchanges under 3 ms. A threat model highlights residual risks such as side-channel leakage and physical tampering.

**Weaknesses:**

1. Validation covers only one ARM/AMD pairing; no results for SGX, Keystone or other emerging platforms.
2. The compliance server is a potential single point of failure, and its scalability is untested.
3. End-to-end latency on hospital Wi-Fi and other constrained networks is not measured.
4. Side-channel defences are described conceptually but lack quantitative evaluation.
5. Physical-access threats like SD-card swaps are deferred to future work despite their prevalence in clinical settings.
6. No comparison with secure multiparty computation or homomorphic encryption, so cost–benefit remains qualitative.
7. Regulatory discussion is high-level; concrete certification pathways and governance engagement are missing.
8. Software-update and key-rotation strategies are not detailed, risking future attestation drift.

---

### Official Review · Reviewer_98g3 · 2025-07-15
**Leveraging Trusted Execution Environments For Data Security in Healthcare Workflows**

**Confidence:** 5
**Clarity Of Writing:** good
**Clinical Significance:** good
**Methodological Novelty:** good
**Overall Rating:** 5

**Experiments And Results:**

good

**Questions For The Authors:**

Dear Authors,

Please see below my questions:

- How do you protect the compliance server? Is it replicated or made tamper-evident? What’s your recovery model if it fails or is attacked?
- Can you elaborate on how easily the prototype scales across more devices and institutions, and what work remains to turn it into a production-ready system?
- Can you explain the effectiveness of the software-only security layers compared to hardware-backed TEEs in your prototype?

Good luck with your paper.

Best Regards

**Strengths:**

- **Practical, End-to-End Security:** Demonstrates a working prototype that ensures confidentiality, integrity, and compliance from data capture to cloud inference, which is rare in healthcare AI security research.
- **Hardware-Agnostic Architecture:** Proposes a flexible framework that works across different TEEs (e.g., TrustZone, SEV, SGX), addressing interoperability challenges in heterogeneous healthcare environments.
- **Low Overhead:** Benchmarks show the added security incurs only ~3% latency, making the solution viable for time-sensitive clinical workflows.
- **Data-Centric Policy Enforcement:** Introduces a secure data tunnel that incorporates provenance, policy tagging, and automated attestation, enhancing auditability and compliance with regulations like HIPAA and GDPR.
- **Reduction of Trusted Base:** By embedding compliance and anonymization within TEEs, the need for intermediate trusted personnel (e.g., data engineers) and infrastructure is reduced, decreasing the attack surface.

**Summary Of The Paper:**

The paper proposes a unified, hardware-agnostic security framework to protect sensitive data in biomedical AI workflows by leveraging Trusted Execution Environments (TEEs) such as ARM TrustZone and AMD SEV. Addressing limitations in interoperability and inconsistent threat models across heterogeneous devices from edge sensors to cloud servers, it introduces a software-defined secure data tunnel that enforces encryption, mutual attestation, access control, and policy compliance. A prototype combining a TrustZone-enabled Raspberry Pi with an SEV-enabled cloud VM demonstrates end-to-end confidentiality and integrity with minimal latency overhead (~3%), making the approach suitable for real-time clinical applications like ARDS detection and scalable to larger workflows such as autism risk prediction from video.

**Weaknesses:**

- **Limited Physical Security:** The paper does not address physical tampering or side-channel attacks on devices like Raspberry Pi, which remain vulnerable in real-world deployments.
- **Deployment Complexity:** Integrating and maintaining multiple TEEs with custom attestation and compliance frameworks across diverse hardware may be operationally complex and resource-intensive.
- **Limited Real-World Evaluation:** The prototype is promising but has been tested on controlled setups; broader deployment across real clinical settings is not yet demonstrated.
- **Assumes Trust in Compliance Server:** The compliance server is a central point for attestation and policy enforcement; if compromised, it could undermine the security of the entire pipeline.

---

### Official Review · Reviewer_9AqP · 2025-07-18
**Hardware-Agnostic TEE Integration for Healthcare Shows Promise But Lacks Rigorous Evaluation**

**Confidence:** 4
**Clarity Of Writing:** great
**Clinical Significance:** good
**Methodological Novelty:** poor
**Overall Rating:** 4
**Final Rating:** 6

**Experiments And Results:**

fair

**Questions For The Authors:**

How do you justify the sample size of 10 runs for performance claims?

What formal security analysis has been conducted? Can you provide results from penetration testing, side-channel analysis, or formal verification of the security properties claimed?

How does your approach compare quantitatively to existing healthcare security frameworks or alternative TEE solutions? What specific advantages does your hardware-agnostic monitor provide over other frameworks?

Can you provide sufficient implementation details for reproduction, including the compliance server architecture, policy specification language, and attestation protocol details

What is the economic feasibility for resource-constrained healthcare settings?

**Strengths:**

The paper tackles a genuine interoperability challenge in healthcare TEE deployment, motivated by concrete use cases (ARDS ventilator monitoring, autism risk detection from video).

The integration of ARM TrustZone at the edge with AMD SEV in the cloud represents a realistic deployment scenario that bridges resource-constrained medical devices with cloud infrastructure.

The paper provides a systematic analysis of vulnerabilities across the healthcare AI pipeline, and Table II maps specific attacks to mitigation strategies.

The data-centric policy enforcement and provenance tracking directly address HIPAA and healthcare regulatory requirements, which is crucial for real-world adoption.

The approach spans from bedside data acquisition through cloud analytics to external research access, covering the complete clinical data lifecycle.

**Summary Of The Paper:**

This paper proposes a hardware-agnostic security architecture that integrates heterogeneous Trusted Execution Environments (TEEs) for end-to-end data protection in biomedical AI workflows. The authors address the challenge that different TEE implementations (ARM TrustZone, Intel SGX, AMD SEV) have incompatible threat models, preventing seamless data protection across edge-to-cloud pipelines.

Their solution consists of two main components: (1) a hardware-agnostic security monitor that orchestrates world/VM switches and provides uniform attestation APIs across different TEEs, and (2) a software-defined secure tunnel that enforces data-centric policies, provenance tracking, and compliance logging. The approach is demonstrated through an ARDS detection pipeline using a TrustZone-enabled Raspberry Pi at the edge connected to an AMD SEV virtual machine in Google Cloud. Performance evaluation shows 3% latency overhead across 10 experimental runs, and attestation overhead measurements are provided for different data transfer scenarios.

**Weaknesses:**

While a threat model is provided, there is no formal security analysis, penetration testing, or evaluation of side-channel attacks. The system's security guarantees are asserted rather than proven. Physical tamper resistance and microarchitectural attacks are acknowledged but not addressed.

No comparison with existing healthcare security frameworks, alternative TEE orchestration approaches, or traditional security measures. This makes it impossible to assess the relative merit of the proposed approach.

Critical implementation details are missing, no code availability is mentioned, and the compliance server implementation is not described in sufficient detail for replication.